

**1** **Ammonium and nitrate additions differentially affect soil microbial biomass of different**

**2** **communities and enzyme activities in slash pine plantation in subtropical China**

Chuang Zhang[a,b], Xin-Yu Zhang[b,c], Hong-Tao Zou[a], Liang Kou[b], Yang Yang[b,d], Xue-Fa Wen[b,c],
Sheng-Gong Li[b,c], Hui-Min Wang[b,c] Xiao-Min Sun[b,c]
[a]College of Land and Environment, Shenyang Agricultural University, Shenyang 110866, China;
[b]Key Laboratory of Ecosystem Network Observation and Modeling, Institute of Geographic
Sciences and Natural Resources Research, Chinese Academy of Sciences, Beijing 100101, China;
[c]College of Resources and Environment, University of Chinese Academy of Sciences Beijing,
100190, China
[d]College of Geographic Science, Harbin Normal University, Harbin 150025, China
Corresponding author: X. Y. Zhang (zhangxy@igsnrr.ac.cn), H.T. Zou
(zouhongtao2001@163.com)

**15** **Abstract**

The ratios of nitrate to ammonium in wet atmosphere nitrogen (N) deposition compounds were
increasing recently. However, the individual effects of nitrate and ammonium deposition on soil
microbial biomass of different communities and enzyme activities are still unclear. We conducted a
four-year N addition field experiment to evaluate the responses of soil microbial biomass of different
communities and enzyme activities to ammonium ($NH_4Cl$) and nitrate ($NaNO_3$) additions. Our results
showed that (1) the inhibitory effects of ammonium additions on total mass of phospholipid fatty acid
(PLFA) were stronger than those of nitrate additions. Both decreased total PLFA mass about 24% and
11% across three sampling time, respectively. The inhibitory effects of ammonium additions on gram
positive bacteria ($G^+$) and bacteria, fungi, actinomycetes (A), and arbuscular mycorrhizal fungi (AMF)
PLFA contents ranged from 14%- 40% across three sampling time. (2) Both ammonium and nitrate
additions inhibited absolute activities of C, N, and P hydrolyses and oxidases, and nitrate additions had
stronger inhibition effects on the acid phosphatase (AP) than ammonium additions. Both ammonium
and nitrate additions decreased N-acquisition specific enzyme activities (enzyme activities normalized
by total PLFA mass) about 21% or 43%, respectively. However, ammonium additions increased





P-acquisition specific enzyme activities about 19% comparing to control. (3) Redundancy analysis
(RDA) showed that the measured C, N, and P hydrolyses and polyphenol oxidase (PPO) activities were
positively correlated with soil pH and ammonium contents, but negatively with nitrate contents; the
mass of PLFA biomarkers were positively correlated with soil pH, soil organic carbon (SOC), and total
N contents, but negatively with ammonium contents. (4) The soil enzyme activities varied seasonally in
the order of March > June > October. On the contrary, microbial PLFA mass was higher in October
than in March and June. Our results concluded that inhibition of mass of PLFA biomarkers and enzyme
activities might be contributed to acidification caused by ammonium addition. Soil absolute enzyme
activities were inhibited indirectly by acidification and nitrification, but specific enzyme activities
normalized by PLFA were directly affected by N additions. It was meaningful to separate the effects of
ammonium and nitrate additions on soil microbial communities and enzyme activities.

**Copyright statement**

The work described has not been published before; it is not under consideration for publication

elsewhere; and its publication has been approved by all the authors and by the responsible authorities of
the institutes where the work was carried out. Any material that has already been published or
copyrighted elsewhere could be reproduced. The copyright of this article is retained by the authors.
Authors grant Copernicus Publications a licence to publish the article and identify itself as the original
publisher. Authors grant Copernicus Publications commercial rights to produce hardcopy volumes of
the journal for purchase by libraries and individuals. Authors grant any third party the right to use the
article freely under the stipulation that the original authors are given credit and the appropriate citation
details are mentioned.

**1. Introduction**

Wet atmospheric nitrogen (N) deposition had increased 25% in the past decade (Jia et al., 2014),

which caused a series of problems in forest ecosystems, such as induced soil acidification, aggravated
the leaching of cation and nitrate, and decreased microbial biomass (Liu et al., 2011; Huang et al., 2014;
Gao et al., 2015; Liu et al., 2013). Although most of wet atmospheric N deposition was ammonium,



nitrate had improved over years, which the ratio of ammonium to nitrate decreased from 5 to 2 (Liu et
al., 2013). Study the differential effects of the two forms of N additions on soil microorganisms could
improve our ability to predict the cycling of C, N and P under nitrate deposition increasing scenario.
Soil microorganism supplies nutrients to forests by producing enzymes to catalyze the degradation of
soil organic matter, drives the cycling of carbon (C), nitrogen (N) and phosphorus (P), and therefore,
influences the forest productivity and sustainability (Heijden et al., 2008). Soil microbial biomass of
different communities could be quantified by phospholipid fatty acid (PLFA) biomarkers. Although the
use of PLFA signature to evaluate microbial diversity was not as advanced as genomics technology,
PLFA method was widely applied to analyze the biomass and microbial community structures
(Frostegård et al., 2011). Usually, bacteria (B), including gram positive ($G^+$) and negative ($G^-$) bacteria,
are liable to degrade labile compound by excreting hydrolase. And fungi (F), including arbuscular
mycorrhizal fungi (AMF) and saprophyte (SAP), are liable to degrade recalcitrant compound by
secreting oxidase (Burns et al., 2013; Sinsabaugh et al., 2010; Willers et al., 2015).
However, only few field studies reported individual effects of ammonium and nitrate additions on
microbial communities in forest soils. Most studies paid more attention to the influence of organic N to
microbial communities (Guo et al., 2010; Hobbie et al., 2012). Compared to nitrate, ammonium with
positive charge could be more easily absorbed by soil colloid with negative charge. Thus, ammonium
would be more available to microorganism than nitrate. However, the process of nitrification, i.e.
ammonium transforming rapidly to nitrate when entering into soil, would sterilize microorganisms in
soil (Dail et al., 2001). There were mechanisms caused different effects of ammonium and nitrate
additions on soil microbial biomass of different communities and enzyme activities. Ammonium and
nitrate additions had different effects on microbial decomposition rate and microbial respiration of soil
organic matter. For example, ammonium additions increased substrate respirations, while nitrate
additions had no influence on substrate respirations in peatland (Currey et al., 2010); Nitrate additions
had strong promotion effects on the decomposition rate of soil organic matter for fir plantation in the
early incubation phase (0-15d; Zhang et al., 2012). However, the inhibition effect of nitrate additions
on soil microbial respiration was similar to ammonium additions in a laboratory incubation experiment
(Ramirez et al., 2010). It was unclear whether ammonium and nitrate additions had a different
influence on soil microbial biomass of different communities.
It was well known that microorganisms and enzymes were sensitive to soil pH. A meta analysis of



soil acidification caused by N additions suggested that ammonium nitrate ($NH_4NO_3$) additions
contributed more to soil acidification than ammonium additions (Tian and Niu, 2015). Most studies did
not differentiate the individual effects of nitrogen addition forms on PLFAs and MBC in forest
ecosystems. A meta analysis reported that N additions decreased MBC by 15%, and fungi were more
sensitive to N additions than the other microbial communities (Treseder et al., 2008). A wide range of
factors could influence the response of microbial biomass to nitrogen additions, including forest type
and geographical location. For example, in temperate forests, $NH_4NO_3$ additions decreased microbial
total PLFAs contents in American beech (*Fagus grandifolia* Ehrh) and yellow birch (*Betula*
*alleghaniensis* Britton), but increased in eastern hemlock (*Tsuga Canadensis* (L.) Carr) and red oak
(*Quercus rubra* (L.) Britton), and the responses of bacteria and fungi were variable (Weand et al.,
2010). In subtropical forest, $NH_4NO_3$ additions increased microbial total PLFAs contents in Chinese fir
(Dong et al., 2015), but decreased soil MBC contents in evergreen broad leaved forests, and no
influence on the pine broad-leaved mixed forest (Wang et al., 2009). To date, the effects of N on soil
microbial communities were inconsistent and it was still unclear how ammonium and nitrate additions
influenced microbial communities, individually.
Soil enzymes catalyze the decomposition of soil organic matter (Burns et al., 2013). The common
labile C-degradation enzymes included α-1,4-glucosidase (αG), β-1,4-glucosidase (βG),
cellobiohydrolase (CBH) and β-1,4-xylosidase (βX) that can decompose starch, cellulose and
hemicellulose. Nitrogen-degradation enzyme includes β-1,4-N-acetylglucosaminidase (NAG) that can
decompose oligosaccharides. Phosphorus-degradation enzyme included acid phosphatase (AP) that can
decompose chitin lipophsphoglycan (Stone et al., 2014). Recalcitrant C-degradation enzymes included
peroxidase and phenol oxidase that can decompose lignin, aromatic and phenolic compounds
(Sinsabaugh et al., 2010). While few study reported the differential effects of ammonium and nitrate
additions on soil enzyme activities in forest ecosystem. In other ecosystem, eg. peatland and alpine
meadow, it showed different effect (Currey et al., 2010; Tian et al., 2014). For example, ammonium
and nitrate additions had an obvious different effect on carbon-, phosphorus-enzyme activities (CBH,
AP) but not for PPO in peatland (Currey et al., 2010). While no significant effects were found in alpine
meadow (Tian et al., 2014).
According to the economic theory, the microorganisms will allocate enzymes to the resources that
were absent for microorganisms, thus N additions relatively increased C, P-acquisition enzymes and





decrease N-acquisition enzymes (Burns et al., 2013). However, a meta analysis reported that N
additions without considering inorganic N forms not only increased the C-degradation enzymes (αG,
βG, CBH and βX) and P-degradation enzymes (AP), and restricted oxidase (PPO and PER), but did not
inhibited N-degradation enzymes (NAG) (Jian et al., 2016; Marklein and Houlton, 2012). It suggested
that allocation of enzyme activities did not completely follow the economic theory.
The response of enzyme activities to N additions were influenced by a series of factors including
environmental conditions, plant types and N background values. For example, in temperate region, the
soil activities of BG, CBH, NAG and PPO were improved by $NH_4NO_3$ additions in dogwood forest, but
were decreased in oak, and were invariant in maple forest. The AP activities were increased in
dogwood and maple forests, but were invariant in oak forest response to $NH_4NO_3$ additions
(Sinsabaugh et al., 2002). However, in acidification temperate region, $NH_4NO_3$ additions increased soil
BG activities in maple forest, but had no influence on soil BG, NAG and AP in yellow birch, oak,
hemlock and beech forests (Weand et al., 2010). In subtropical and tropical forests, $NH_4NO_3$ additions
increased BG, NAG, AP activities, and decreased oxidase (PPO and PER) activities (Dong et al., 2015;
Guo et al., 2011; Cusack et al., 2011). In general, it was still unclear how N addition affected on
enzyme activities and whether there were different effects of ammonium and nitrate additions on
enzyme activities. To better predict the effect of elevated N deposition on soil cycling of C, N, and P, it
was necessary to evaluate the individual effects of ammonium and nitrate additions on soil microbial
biomass of different communities and enzyme activities.
The subtropical soils were thought to be N-rich and undergone increasing nitrate deposition in
southern China. We established a long-term nitrate and ammonium additions experiment in the slash
pine (*Pinus elliottii*) plantations in subtropical area. We aimed to explore the differential effects of
ammonium and nitrate on soil microbial communities and enzyme activities, respectively. We
hypothesized that (1) ammonium additions would have a stronger inhibitive ability to total PLFAs,
fungi PLFA contents, and enzyme activities due to its strong negative effect on soil pH; and (2)
ammonium and nitrate additions would increase C, P-hydrolase, but decreased N-hydrolase activities
according to the economic theory, and inhibited oxidase activities due to their effects on fungi.

**2. Materials and methods**




**2.1. Study site**

The study was conducted in Qianyanzhou (QYZ) Experimental site in hilly red soil region, Taihe
country, Jiang Xi province (26°44′29.1″N, 115°03′29.2″E, 102 m a. s. l). The region was subtropical
monsoon climate with mean annual temperature and precipitation of 17.9 °C and 1475 mm,
respectively. Atmospheric wet N deposition was about 33 kg N ha$^{-1}$ yr$^{-1}$ consisting of 11 kg N ha$^{-1}$ yr$^{-1}$
ammonium and 8 kg N ha$^{-1}$ yr$^{-1}$ nitrate, respectively (Zhu et al., 2014). The soil is weathered from red
sandstone and mud stone, and is classified as Typical Dystrudepts Udepts Inceptisols according to US
soil taxonomy (Soil Survey Staff, 2010). The slash pine (*Pinus elliottii*) was planted in 1985 and was
one of the dominant species using vegetation restoration in this hilly red soil region. The dominant
understory vegetation is *Woodwardia japonica*, *Dicranopteris dichotoma* and *Loropetalum chinense*
(Kou et al., 2015).

**2.2. Experimental design**

As described by Kou et al. (2015), the plots were established in November 2011 using a randomized
complete block design. There were two forms of N treatments, i.e. ammonium additions (N$_{ammonium}$)
using ammonium chloride (NH$_4$Cl) and nitrate additions (N$_{nitrate}$) using sodium nitrate (NaNO$_3$), with a
dosage of 40 kg N ha$^{-1}$ yr$^{-1}$ and a Control (CK). Each treatment had three replicates, totally nine plots
(20 m × 20 m, slope <15°). The plots were separated with more than 10 m buffer zone between plots.
The NH$_4$Cl or NaNO$_3$ were dissolved in 30 L tap water and evenly sprayed onto the plots once per
month, i.e. 12 times per year. The equivalent amount of tap water was sprayed onto each Control plot.
Nitrogen additions started on 01-May-2012 and proceeded at a month interval on non-rainy days, and
totally 140 kg N ha$^{-1}$ was inputted when soils were collected.

**2.3. Sampling and analysis**

We collected soil samples at March, June and October of 2015 after removing surface litters, and
mixed 5 cm diameter cores from five randomly selected locations together as one composite sample.
Soil samples were taken from 0-10 cm depth from each plot, then field- fresh samples were sieved




through a 2 mm sieve after mixed evenly. Samples were kept at 4 °C for PLFA biomarkers, enzyme
activities, soil pH, ammonium and nitrate, and soil dissolved organic carbon (DOC) analyses. The
assays of PLFA biomarkers and enzyme activities were performed at once after back to laboratory. A
subsample was air dried, and then sieved through a 0.25 mm mesh for soil organic C (SOC), total N
(TN), and Total P (TP) analyses.
Soil pH was measured in a 1g fresh soil : 2.5 v:v soil-water suspension by glass electrode. Fresh soils
were extracted by $1 mol L^{-1}$ KCl, shaken for 2 hours, and measured by a continuous flow auto-analyzer
(Bran Lubbe, AA3, Germany) to determine ammonium and nitrate contents. Soil DOC was extracted
with 1:5 (v:v) soil : distilled water, and measured by Liquid TOC*II* (Elementar, Germany). Soil TN and
SOC were measured by CN Analyzer (Vario Max, Elementar, Germany).
Phospholipid fatty acid (PLFA) biomarkers were measured according to the methods of Bossio and
Scow (1998). In brief, field-fresh soil equal to 8 g dry soil was undergone mild alkaline methanolusis to
form fatty acid methyl eaters (FAMEs). Then the extraction of PLFA dissolved in hexane was measured
by Agilent 6890N Gas Chromatograph, with MIDI peak identification software (version 4.5; MIDI Inc.
Newark, DE) with a DB-5 column. The abundances of PLFA biomarkers were calculated as nmol
PLFA $g^{-1}$ dry soil. Total amount of the different PLFA biomarkers were used to represent differnt
groups of soil microorganisms, i.e. gram-positive bacteria ($G^+$) by i14:0, i15:0, a15:0, i16:0, i17:0,
a17:0; gram-negative bacteria ($G^-$) by 16:1ω7c, cy17:0, 18:1ω7c, cy19:0; arbuscular mycorrhizal fungi
(AMF) by 16:1ω5; saprophytic fungi (SAP) by 18:1ω9c, 18:2ω6c, 18:2ω9c 18:3ω6c; actinomycete (A)
by 10Me16:0, 10Me17:0, 10Me18:0 (Bradley et al., 2007; Denef et al., 2009). Bacteria biomass were
calculated as the sum of $G^+$ and $G^-$, and fungi biomass were calculated as the sum of AMF and SAP,
respectively.
We measured four C-acquisition hydrolase (i.e. αG, βG, CBH and βX), one N-acquisition hydrolase
(NAG) and one P-acquisition hydrolase (AP) following the methods of Saiya-Cork et al. (2002). Their
corresponding substrates and functions see Table 1. In brief, 1 g field-fresh soil was homogenized in
125 ml of sodium acetate buffer. The buffer was adjusted to 4.5 of pH based on the ambient soil pH.
μl homogenate and 50 μl substrate was added to 96-well black microplates, then incubated at 20 °C
for 4 h. After incubation, 10 μl $1 mol L^{-1}$ NaOH was added to each well to terminate the reactions, and
fluorescence values were measured with 365 nm excitation and 450 nm emission filters by a microplate
fluorometer (Synergy H4,BioTek). Totally, there were eight replicates per soil sample.




Two oxidases (i.e. PER and PPO were measured using 96-well transparent microplates according to
the methods of Saiya-Cork et al. (2002). 600 µl homogenate and 150 µl substrate were added to 96-well
deep microplates. When measuring PER activities, 10 µl of 0.3% $H_2O_2$ was added to homogenate and
substrates mixtures. After incubated at 20 $^{o}$C for 5 h, the microplates were centrifuged at 3000 r for 3
minutes, then transferred 250 µl liquid supernatant to 96-well transparent microplate. Absorbance
values were measured at 460 nm by microplate spectrophotometer (Synergy H4, BioTek). The
corresponding substrates and their functions of the measured enzymes were shown in S 2.
After correcting for homogenate control, substrate control and quenching, absolute activities were
expressed in units of nmol $g^{-1}$ soil $h^{-1}$. We calculated the specific activities of the enzymes by dividing
enzyme activities by PLFA values to normalize the activity to the size of the microbial active biomass
(Cusack et al. 2011).

**2.4. Statistical analyses**

Two factors randomized block variance of analyses and Duncan analyses were applied to test the
differences between treatments and sampling time. One-way analysis of variance (ANOVA) and
Duncan analyses were applied to test the difference of the treatments in individual sampling time.
Analyses were performed using SPSS 17.0. Relationships among the soil physical-chemical properties,
soil PLFA biomarkers and the soil enzyme activities were tested by redundancy analysis (RDA) using
CANOCO 4.5. Statistical significance was determined as $P < 0.05$. The figures were drawn by
sigmaplot 10.0.

**3. Results**

**3.1. Soil physical and chemical properties**

Totally, treatments have a significant influence on soil pH (F=12.43, $P<0.01$), DOC (F=23.53,
P<0.01), nitrate (F=43.19, P<0.01) and ammonium (F=11.84, P<0.01) (Table 2). Ammonium additions
decreased soil pH by 0.7 unit across three sampling time, while nitrate additions did not affect soil pH
significantly. Nitrate additions increased soil DOC by 17% across three sampling time, while



ammonium additions did not affect soil DOC significantly. Ammonium and nitrate additions increased
soil nitrate contents by 165% and 129%, respectively, but they all decreased soil ammonium contents
by 31% and 38% across three sampling time, respectively (Fig.1).
The sampling time have a significant influence on DOC (F= 561.25, P<0.01), nitrate (F=7.96,
P<0.01) and ammonium (F= 65.46, P<0.01), but not on soil pH (Table 2). DOC contents were in order
of March < June < October. In contrast to nitrate contents, ammonium contents were in order of
March > June and October.

**3.2. Soil microbial biomass of different communities**

Both treatments and sampling time had a significant influence on soil microbial biomass of different
communities (P<0.01, Table 2). Totally, ammonium and nitrate additions decreased total PLFAs
contents, and the effects of ammonium additions on the different PLFA biomarkers were stronger than
those of nitrate additions across three sampling time. Ammonium additions decreased total PLFA
contents by 24 %, and decreased $G^+$, AMF, B, F, A PLFA contents by 14 % - 40 % across three
sampling time. Nitrate additions decreased total PLFA contents by 11%, and decreased $G^+$, AMF, B, F,
A PLFA contents by 7% - 24% across three sampling time. Only ammonium additions shifted the
microbial communities from $G^-$ to $G^+$, i.e. increased the ratio of $G^+/G^-$ comparing to CK or nitrate
additions. Additionally, both ammonium and nitrate additions decreased the ratios of F/B, but the
effects of nitrate additions were stronger than those of ammonium additions (Fig.2).
Additionally, the measured soil PLFA biomarkers exhibited seasonal variations (Table 2). Total
PLFA and PLFA contents of B, F, $G^+$, $G^-$, AMF, SAP were in order of March > June > October. PLFA
contents of A were in order of June > March > October.

**3.3. Soil enzyme activities**

Both treatments and sampling time had a significant influence on the measured absolute enzyme
activities (P<0.01, Table 2), i.e. ammonium inhibited by 15%-43% and nitrate by 6%-50% across three
sampling time, respectively (Table 3). The AP absolute activities were about 9% lower under nitrate
than under ammonium additions (Table 3).



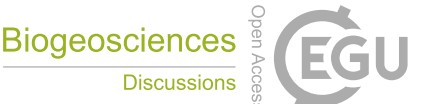

The treatments had a significant influence on N, P-acquisition specific enzyme activities (P<0.01),
but not on C and oxidase specific enzyme activities (Table 2). The inhibition effects of nitrate additions
on the N-acquisition specific enzyme activities (about 43%) were stronger than those under the
ammonium additions (about 21%) across three sampling time. And only ammonium additions
increased the AP specific activities (about 19%) compared to the CK across three sampling time
(Fig.3).
Additionally, the measured enzyme activities exhibited seasonal variations (Table 2). BG, BX, CBH,
NAG, AP and PPO activities were in order of March < June < October. aG activities were in order of
March > October > June, and PER activities were in order of March > June > October (Table 3).

**3.4. Redundancy analyses**

The RDA between soil properties, enzyme activities, and PLFA biomarkers showed that the first
ordination RDA axis explained 72.0% and 66.8%, respectively, the second axis explained 11.5% and
13.2%, respectively. The RD1 for soil enzyme activities and PLFA biomarkers were correlated with
DOC/SOC, DOC, ammonium, and SOC. However, nitrate was only correlated with the RD1 of enzyme
activities but not that of PLFA biomarkers. Most of the measured soil enzyme activities and the PLFA
biomarkers were positively correlated with soil pH, but $G^+/G^-$ and F/B were negatively correlated with
soil pH. Ammonium and DOC were positively correlated with the soil enzyme activities except PER,
but negatively with PLFA biomarkers. Nitrate was negatively correlated with soil enzyme activities, but
hardly with PLFA biomarkers (Fig. 4).

**4. Discussion**

In agreement with our first hypothesis, our results showed that both ammonium and nitrate additions
significantly decreased soil total mass of PLFA biomarkers, bacteria, fungi, actinomycetes, $G^+$, AMF,
SAP-PLFA contents, and ammonium additions had stronger inhibition effects on PLFA biomarkers
across three sampling time (Figure 2, Table 2). Soil microbial biomass was negatively influenced by
resource availability and acidification (Sinsabaugh et al., 2014; Moorhead et al., 2006). However, N
additions tended to increase soil DOC contents, and available N (sum of ammonium and nitrate



contents) did not change in response to N additions in our study. It suggested that PLFA biomarkers
contents were inhibited by some other factors except soil availability of C and N in the subtropical
slash pine (*pinus elliottii*) forest. The RDA analysis showed the positive correlations between PLFA
biomarkers contents and soil pH (Fig. 4). Acidification caused by ammonium additions might be
attributed to decrease of mass of microbial PLFA. Ammonium additions could aggravate nitrification
in subtropical soils (Tang et al. 2016), and nitrification might be toxic to microorganism (Dail et al.,
2001), which would decrease microbial PLFA contents. Nitrate additions had no influence on soil pH
(Fig 1), which would explain why nitrate addition had weaker inhibition effects on mass of PLFA
biomarkers. The possible reasons that nitrate addition inhibited the mass of PLFA biomarkers might be
as follows, nitrate additions could accelerate leaching of $Ca^{2+}$, $Mg^{2+}$ (Qian et al., 2007), increase soil
osmotic potential, and activate $Al^{3+}$ absorbed by soil colloid (Treseder et al., 2008). Additionally, N
additions decreased the PER activity, which would cause polyphenol accumulation in soil.
Accumulated polyphenol might also be toxic to microorganism (Sinsabaugh et al., 2010) and
contributed to decrease the contents of PLFA biomarkers.
In our study, both ammonium and nitrate additions decreased the ratios of fungi /bacteria, suggesting
that fungi were more sensitive to N additions. We found that N additions decreased fine root biomass in
our previous study (Kou et al., 2015), and N additions could destroy symbiotic system between AMF
and plants, so that restrict AMF-PLFA contents.
Our study showed that both ammonium and nitrate additions inhibited the absolute activities of C, N,
P-hydrolase and oxidase across three sampling time (Table 2, Table 3). It agrees with our hypothesis
and the economic theory that N additions decreased the absolute activities of N-acquisition enzyme
(NAG). However, it does not agree with our hypothesis that N additions will increase the C- or
P-acquisition enzymes. We found positive correlations between soil pH, ammonium contents and the
measured enzyme activities except PER, and negative correlations between nitrate contents and most of
the measured enzyme activities (Figure 4), indicating that acidification and nitrification could restrict
enzyme activities. Microorganisms were main producers of soil enzymes, the decrease of microbial
biomass would reduce soil absolute enzyme activities (Allison et al., 2005).
We found that the specific enzyme activities of N, P-acquisition were different after ammonium and
nitrate additions (Figure 3, Table 2). The specific enzyme activities of C-hydrolase and oxidase
maintained constant under N additions, although N additions restricted the absolute activities of



C-acquisition enzymes. Microorganisms tended to preferentially allocate energy resource (C) to meet
their growth demanding (Schimel and Schaeffer, 2012). Nitrate additions had stronger inhibition effects
on the specific enzyme activities of N-acquisition than under ammonium additions. It is indicated that
N addition decreased the N-demanding of unit-microbial biomass. Analogously, increase of
P-acquisition specific enzyme activities under ammonium additions suggests the increase of
P-demanding of unit-microbial biomass in the P-limited subtropical region. Acidification due to
ammonium additions might aggravate P-deficiency by reactivating $Al^{3+}$ reaction with available P
(Vitousek et al., 2010; Mohren et al., 1986), which would improve the demanding of P. Additionally,
soil absolute enzyme activities would be influenced more strongly by abiotic, i.e. soil pH, than biotic
conditions (Kivlin et al., 2016). Declines of C, P-acquisition absolute enzyme activities might be
attributed to the edaphic variations such as acidification and nitrification.
We also found significant seasonal variations in mass of PLFA biomarkers and enzyme activities
(Table 2). Microbial PLFA contents were higher in October, which may be explained by litter increase
in October. Fresh litter inputs could promote decomposition of old recalcitrant compounds
(Blagodatskaya and Kuzyzakov, 2008), which might be confirmed by the high PER activities in
October. Additionally, we also found there were interactive effects of N treatments and different
sampling time on soil enzyme activities and PLFA contents of biomarkers (Table 2). It suggested that
soil microbial biomass and enzyme activities were simultaneously influenced by a series of factors,
such as atmosphere conditions, precipitation, and sequent change of soil variables.

**5. Conclusions**

The results showed that both ammonium and nitrate additions decreased soil total microbial PLFA
mass, and PLFA mass of bacteria, fungi, actinomycetes, $G^+$, $G^-$, AMF, SAP. The inhibitive effects on
the biomass of different soil microbial communities except SAP were more significant under
ammonium additions than under nitrate additions. It might be attributed to acidification caused by
ammonium additions since PLFA biomarkers were positively correlated with pH. Ammonium additions
shifted microbial communities to $G^+$ and bacteria-dominated, and nitrate additions shifted microbial
communities to bacteria-dominated.
Although ammonium and nitrate additions reduced absolute enzyme activities of C, N, and



P-acquisition, the specific enzyme activities of P-acquisition were increased under ammonium
additions, and specific enzyme activities of C-acquisition maintained constant. It suggested that
ammonium and nitrate additions increased the microbial demanding of C and P. Soil absolute enzyme
activities were inhibited indirectly by acidification and nitrification, but specific enzyme activities
normalized by PLFA were directly affected by N additions.
In general, the effects of ammonium and nitrate additions on soil microbial communities and specific
enzyme activities was various. In order to better predict the elevated N deposition on soil microbial
functions and enzyme activities, it was necessary to discuss the effect of ammonium and nitrate,
separately.

*Author contribution:* Xin-yu Zhang, Xue-Fa Wen, Sheng-Gong Li, Hui-Min Wang, and Xiao-Min Sun
designed research; Liang Kou performed research; Chuang Zhang, Yang Yang and Xin-yu Zhang
analyzed data; and Chuang Zhang wrote the paper.

*Competing interests:* The authors declare no conflict of interest.

*Acknowledgments*

This study was jointly financed by the Major, State Key and General Programs of National Natural
Science Foundation of China (Nos. 31130009, 41571130043, 41571251)

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





**Figure Legends**

**Fig. 1.** The effects of ammonium and nitrate additions on soil pH, DOC, nitrate and ammonium
contents in individual sampling time. Capital letters represent significant differences between the
treatments ($P$ <0.05), and small letters represent significant between the different sampling time ($P$
<0.05). Error bars represent standard errors, the same below.
**Fig. 2.** The effects of ammonium and nitrate additions on PLFA biomarkers in different sampling time.
**Fig. 3.** The effects of ammonium and nitrate additions on C, N, P-acquisition specific enzyme and
oxidase specific activities in different sampling time.
**Fig. 4.** Redundancy analyses between (a) soil properties and enzyme activities; (b) soil properties and
PLFA-biomarkers.






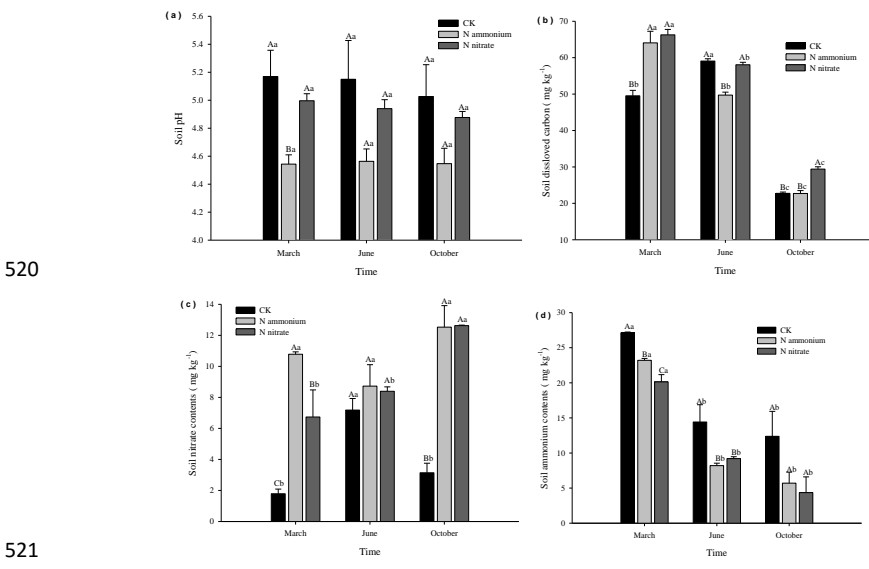

**Fig.1**








**Fig. 2**






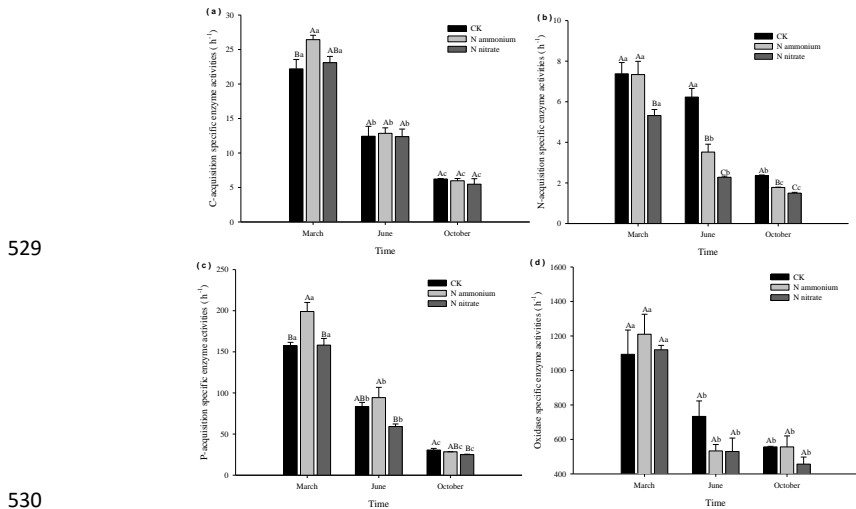

**Fig.3**



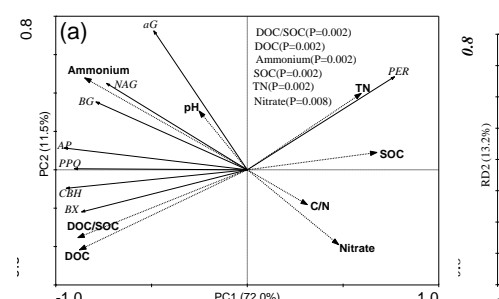
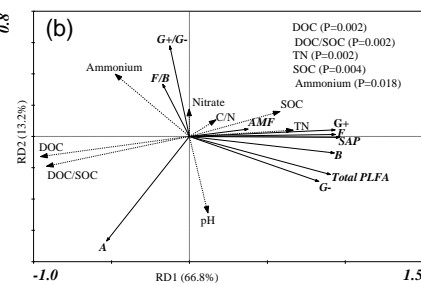

**Fig. 4**



**Table 1** Enzymes and their corresponding substrates and functions

| Enzyme | Ec | Abbreviation | Substrate | Function |
|---|---|---|---|---|
| Peroxidase | 1.11.1.7 | PER | L-DOPA | Oxidize lignin and aromatic compounds using $H_2O_2$ or secondary oxidants as an electron acceptor |
| Phenol oxidase | 1.10.3.2 | PPO | L-DOPA | Oxidize phenolic compounds using oxygen as an electron acceptor |
| α-1,4-glucosidase | 3.2.1.20 | aG | 4-MUB-α-D-glucoside | Releases glucose from starch |
| β-1,4-glucosidase | 3.2.1.21 | BG | 4-MUB-β-D-glucoside | Releases glucose from cellulose |
| Cellobiohydrolase | 3.2.1.91 | CBH | 4-MUB-β-D-cellobioside | Releases disaccharides from cellulose |
| β-1,4-xylosidase | 3.2.1.37 | BX | 4-MUB-β-D-xyloside | Releases xylose from hemicellulose |
| β-1,4-N-acetylglucosaminidase | 3.2.1.14 | NAG | 4-MUB-N-acetyl-β-D-glucosaminide | Releases N-acetyl glucosamine from oligosaccharides |
| Acid phosphatase | 3.1.3.1 | AP | 4-MUB-phosphate | Releases phosphate groups |




**Table 2** Summary statistics (F ratio, P value) for two factors randomized block variance of analyses (ANOVA) and Duncan analyses
applied to soil variables, enzyme activities and PLFA biomarkers. P value that are significant level ($P < 0.05$)

| Factors | Treatments | Months | Treatments × Months |
|---|---|---|---|
| pH | **12.43, 0.00** | 0.31, 0.74 | 0.09 , 0.99 |
| DOC | **23.53, 0.00** | **561.25, 0.00** | **20.11, 0.00** |
| Nitrate | **43.19, 0.00** | **7.96, 0.00** | **8.21, 0.00** |
| Ammonium | **11.84, 0.00** | **65.46, 0.00** | 0.42, 0.79 |
| TPLFA | **102.51, 0.00** | **477.77, 0.00** | 2.68, 0.07 |
| B | **56.94, 0.00** | **555.14, 0.00** | 2.73, 0.07 |
| F | **180.49, 0.00** | **277.81, 0.00** | **52.16, 0.00** |
| A | **172.23, 0.00** | **2627.61, 0.00** | **123.12, 0.00** |
| G$^+$ | **50.30, 0.00** | **1221.19, 0.00** | **14.39, 0.00** |
| G$^-$ | **34.33, 0.00** | **105.59, 0.00** | 0.45, 0.77 |
| AMF | **147.77, 0.00** | **83.55, 0.00** | **21.64, 0.00** |
| SAP | **24.70, 0.00** | **781.67, 0.00** | **13.08, 0.00** |
| G$^+$/G$^-$ | **16.24, 0.00** | 2.38, 0.12 | 0.94, 0.46 |
| F/B | **3.82, 0.04** | **56.42, 0.00** | **21.67, 0.00** |
| aG | **30.24, 0.00** | **53.17, 0.00** | **3.47, 0.03** |
| BG | 3.26, 0.07 | **72.90, 0.00** | 0.58, 0.68 |
| BX | **9.86, 0.00** | **79.08, 0.00** | **3.86, 0.02** |
| CBH | **28.51, 0.00** | **194.75, 0.00** | **4.39, 0.01** |
| NAG | **100.42, 0.00** | **67.49, 0.00** | **8.47, 0.00** |
| AP | **22.81, 0.00** | **467.77, 0.00** | 1.73, 0.19 |
| PPO | **6.87, 0.01** | **64.40, 0.00** | 1.98, 0.15 |
| PER | **6.27, 0.01** | **194.30, 0.00** | **3.07, 0.05** |
| C-acquisition specific enzyme | 2.82, 0.09 | **334.41, 0.00** | 2.07, 0.13 |
| N-acquisition specific enzyme | **29.10, 0.00** | **128.31, 0.00** | **6.36, 0.00** |
| P-acquisition specific enzyme | **13.42, 0.00** | **397.19, 0.00** | **4.53, 0.00** |
| Oxidase specific enzyme | 1.68, 0.22 | **89.04, 0.00** | 1.84, 0.17 |





**Table 3** Summary statistics (mean ± standard error) for One way analyses (ANOVA) and Duncan analyses applied to soil absolute enzyme activities. Capital letters represent significant differences between the treatments ($P <0.05$), and small letters represent significant between the different sampling time ($P <0.05$).

| Months | Treatments | aG nmol g⁻¹ h⁻¹ | BG nmol g⁻¹ h⁻¹ | BX nmol g⁻¹ h⁻¹ | CBH nmol g⁻¹ h⁻¹ | NAG nmol g⁻¹ h⁻¹ | AP nmol g⁻¹ h⁻¹ | PPO μmol g⁻¹ h⁻¹ | PER μmol g⁻¹ h⁻¹ |
|---|---|---|---|---|---|---|---|---|---|
| March | CK | 7.0±0.1 Aa | 160.9±15.6 Aa | 36.4±3.4 Aa | 30.±2.1 Aa | 77.5±4.7 Aa | 1658.7±59.1 Aa | 7.9±0.9 Aa | 1.4±0.1 Ab |
| | N ammonium | 4.5±0.2 Ba | 143.5±4.0A a | 26.8±3.2 Aa | 27.3±1.5 Aa | 56.1±5.2 Ba | 1520.7±78.2 Aa | 8.9±0.0 Aa | 1.5±0.1 Ab |
| | N nitrate | 4.5±0.2 Ba | 157.1±10.9 Aa | 33.4±1.0 Aa | 21.0±0.8 Ba | 49.7±2.6 Ba | 1475.2±53.2 Aa | 9.9±1.4 Aa | 1.6±0.1 Ab |
| June | CK | 4.0±0.9 Ab | 83.2±13.0A b | 37.2±1.6 Aa | 28.6±2.5 Aa | 77.0±4.7 Aa | 1030.3±41.2 Ab | 7.7±1.2 Aa | 1.4±0.1 Ab |
| | N ammonium | 2.2±0.1 ABc | 70.6±0.9 Ab | 25.9±1.8 Ba | 17.9±0.2 Bb | 31.8±1.7 Bb | 848.5±62.1 Bb | 4.0±0.0 Bb | 0.9±0.1 Bb |
| | N nitrate | 1.7±0.3 Bb | 89.4±10.3A b | 28.7±1.2 Bb | 19.8±0.2 Ba | 25.7±0.6 Bb | 667.8±26.5 Cb | 4.8±0.9 ABb | 1.2±0.1 Ab |
| October | CK | 3.7±0.4 Ab | 89.1±0.9 Ab | 15.2±0.4 ABb | 9.7±0.3A b | 44.7±0.2 Ab | 578.0±38.1A c | 2.9±0.2 Ab | 7.6±0.1 Aa |
| | N ammonium | 3.7±0.1 Ab | 64.0±4.2 Ab | 16.2±0.9 Ab | 5.2±0.1 Bc | 26.5±0.2 Bb | 423.4±1.6 Bc | 2.8±0.1 Ab | 5.5±0.8 Aa |
| | N nitrate | 2.2±0.0 Bb | 68.3±11.5A b | 13.5±0.1 Bc | 5.3±0.1 Bb | 24.5±0.2 Cb | 409.8±4.7 Bc | 1.9±0.1 Bc | 5.6±0.8 Aa |
