# Peer review of "Contrasting effects of ammonium and nitrate additions on the biomass of soil microbial"

_Biogeosciences, 2017_

## Referee Comment (RC1) · Anonymous Referee #1 · 20 Jun 2017

The effect of atmospheric nitrogen deposition on forest ecosystems has become one important issue and popular research topic in recent decades across the world. The aim of this manuscript is to explore the differential effects of ammonium and nitrate on soil microbial communities and enzyme activities. The ideas are interested and worth to do. In general, I think the manuscript could be published after taking the following questions.

1. The data analyses is not enough or suitable. 1) The authors mentioned that using two factors randomized block variance of analyses (ANOVA) to test the differences

between the treatment and the sampling season. However, after reading the whole manuscript, I did not find any results from this methods. For example, if there is two factors, it must have the possible interaction effect between two factors. Actually, from Table 2, I have found several important interaction effects for DOC, Nitrate, Fugal et al. However, the authors did not discuss this at all. 2) Another question is that I have some confused why the authors took three measurements for PLFA biomarkers, even we knew that the variance of PLFA measurements varied a lot. Is there any important reasons to choose these three different months and what the ecological meanings are? evenly sprayed onto the plot once per month. Did this mean that the frequency of spray the nitrogen 12 times per year?

Other minor comments: 1. In introduction and discussion section: the authors cited papers in different ways, please format it. 2. The authors should improve their English grammar for a lot. There are a lot of small mistakes in the whole manuscript.

---

## Referee Comment (RC2) · Anonymous Referee #2 · 21 Jun 2017

This manuscript by Zhang et al examines the impact of 4 years of simulated atmospheric N deposition in $\sim$30 year old pine-dominated forest plots. The approach used by the authors is novel in that it differentiates between type of N applied (NH4 vs NO3), which is particularly interesting due to the global stabilization/decline in N deposition as NO3 and the increase in N deposition as NH4.

General Comments:

1) In general, the data are much more complex across time than the authors present.

[Figure]

It would be nice if the results are as clean as suggested in the topic sentence of each discussion paragraph, but it is simply not the case because many of the results are time-dependent. For this reason, the authors need to greatly expand the interpretation and discussion of the treatment x time interaction that is presented in Table 2. Further, to help the reader reason through the data, I think it would be beneficial to collapse the data to not include the 3 sampling times for those factors that do not exhibit a significant treatment x time interaction. For example, Fig 2(i) can be reduced to three bars for control, ammonium N, and nitrate N because there was not a significant interaction.

2) More can be done with soil enzyme data to forward the authors main hypotheses and ideas that are introduced in line 119. For example, enzyme data can be presented as ratios of C acquiring/N acquiring and/or C acquiring/P acquiring. Such analysis will provide a clearer avenue to draw conclusions about whether microbes can alter how resources are allocated to scavenge for nutrients under different conditions.

3) Is there any ecological rationale for the March/June/October time points? What is the climatic variation across these times? Also, I assume soil moisture was measured, and if it was, those data should be presented and included in all analyses (including the RDA). Soil moisture has been shown to be a major driver of microbial community composition.

4) The manuscript needs to be edited for grammar, flow, and word choice. The writing is poor and must be improved significantly in order to be publishable.

Minor comments:

Line 169: Is there rationale for the dose of N applied? Any relation to predictions for future N deposition in the region?

Line 229: How were the 3 sampling times considered for the RDA? Was the RDA ran on data from one of the three sampling times? Or from average data across the sampling times? Given the treatment by time interaction, this point needs clarification.

**BGD**

Line 315: Others have shown that N addition disproportionately effects soil fungi and may stimulate soil bacteria (for example, see doi 10.3389/fmicb.2016.00259 and 10.1128/AEM.01224-14). This dynamic may also help explain the increase in DOC observed with N addition.

Table 2: I think P-values can be removed from this table. Given that significant values are bolded, P-values are redundant and make the table busy for the reader.

[Figure]

---

## Short Comment (SC1) · 1 Aug 2017

Comments in response to Referee 1

The effect of atmospheric nitrogen deposition on forest ecosystems has become one important issue and popular research topic in recent decades across the world. The aim of this manuscript is to explore the differential effects of ammonium and nitrate on soil microbial communities and enzyme activities. The ideas are interested and worth to do. In general, I think the manuscript could be published after taking the following questions. Response: We would like to thank you for the helpful and constructive comments, which further improved the manuscript. We have carefully revised our manuscript to take account of your comments and suggestions. Please find below our responses (blue font) to comments (repeated in an italic font). The page and line numbers mentioned here refer to the latest revision of our manuscript simultaneously submitted with all figures as a single PDF file.

1. The data analyses is not enough or suitable. 1) The authors mentioned that using two factors randomized block variance of analyses (ANOVA) to test the differences between the treatment and the sampling season. However, after reading the whole manuscript, I did not find any results from this methods. For example, if there is two factors, it must have the possible interaction effect between two factors. Actually, from Table 2, I have found several important interaction effects for DOC, Nitrate, Fugal et al. However, the authors did not discuss this at all.

Response: We added the results of interaction effects to the result section. "The soil pH and ammonium contents were either treatment- or time-independent. There were interaction effects between the treatments and the sampling time on the soil DOC and nitrate contents (P<0.01, Table 1)." (line 238-240) "Both the treatment and the time of sampling significantly influenced the soil microbial biomass of the different communities (P<0.01). Total PLFAs, bacteria, G−, and G+/G− were either treatment- or time-independent. There were also interaction effects between treatments on sampling time and fungi, actinomycetes, G+, AMF, SAP, and the fungi/bacteria ratio (Table 1)." (line 252-255) "There were significant influences from both treatment and sampling time on the measured absolute enzyme activities (P<0.01). Activities of BG, AP, and PPO were either treatment- or time-independent, and there were interaction effects between the treatments and sampling time on activities of aG, BX, CBH, NAG, and PER (Table

1)." (line 271-274) We also discussed the possible reasons resulted in the interaction effects at the discussion section. "The N treatments also varied significantly on a seasonal basis and there were interaction effects between N treatments and seasons on the contents of some PLFA biomarkers and enzyme activities (Table 2). Climate conditions, plant growth, the amount of litter returned, and plant-soil-microorganism systems varied across the three seasons. The temperature ranged from 13.5 to 27.6 °C, and precipitation ranged from 88.2 to 176.6 mm, across the three seasons (Fig. S1), and did not limit the growth of microorganisms. The positive relationships between PLFA biomarker contents and soil moisture contents indicate that soil moisture had a strong influence on soil microbial community biomass. There may be interaction effects between plant growth, the mass and quality of litter, plant-microbe competition, and soil nutrient dynamics. For example, compared with the control plots, the soil DOC contents were lower, and soil nitrate contents stayed the same in June (the growing season) in the ammonium treatment, but the soil DOC and nitrate contents were higher in the ammonium and nitrate treatments in March and October (the non-growing season, Fig. 2). This indicates that there was stronger competition between plants and microbes for available C and N in June than in March and October, and that there were interaction effects between plants and microbes on soil C and N availability. This might explain the interaction effects between N additions and seasons on the activities of C and N-acquisition enzymes. The effects of interactions between N additions and season on the AMF PLFA contents, along with available C and N dynamics, may result from plant growth as plant-AMF symbiotic systems may be influenced by fine root biomass." (line 354-371)

2) Another question is that I have some confused why the authors took three measurements for PLFA biomarkers, even we knew that the variance of PLFA measurements varied a lot. Is there any important reasons to choose these three different months and what the ecological meanings are? evenly sprayed onto the plot once per month. Did this mean that the frequency of spray the nitrogen 12 times per year?

Response: "We collected soil samples in March, June, and October of 2015, to represent spring, summer, and fall." (line 169-170) "The atmospheric conditions and plant-derived litters differed between the three seasons, and so indirectly affected the soil microbial biomass and enzyme activities of different communities. We collected soils from three seasons so that we could investigate the synthetic responses of soil microbial biomass and enzyme activities to ammonium and nitrate additions and to obtain improved information to support predictions of the effects of elevated N depositions on C, N, and P cycling." (line 172-177) The frequency of spray the nitrogen was 12 times per year. We showed at "The NH4Cl or NaNO3 were dissolved in 30 L of tap water and evenly sprayed onto the plots once a month, i.e. 12 times per year." (line 161-162)

Other minor comments:

1. In introduction and discussion section: the authors cited papers in different ways, please format it.

Response: Revised as recommended.

2. The authors should improve their English grammar for a lot. There are a lot of small mistakes in the whole manuscript.

Response: We have our revised version manuscript professionally edited by a native English speaker colleague, Dr Deborah Ballantine from the United International College, Beijing Normal University and Hong Kong Baptist University, Zhuhai, Guangdong Province.

Please also note the supplement to this comment:
https://www.biogeosciences-discuss.net/bg-2017-179/bg-2017-179-SC1-supplement.pdf
* * *
[Figure]

[Figure]

[revised manuscript text omitted]
 | 160.9±15.6 Aa | 36.4±3.4Aa | 30.±2.1A a | 77.5±4.7 Aa | 1658.7±59.1 Aa | 7.9±0.9Aa | 1.4±0.1 Ab |
| | N ammonium | 4.5±0.2Ba | 143.5±4.0A a | 26.8±3.2Aa | 27.3±1.5 Aa | 56.1±5.2 Ba | 1520.7±78.2 Aa | 8.9±0.0Aa | 1.5±0.1 Ab |
| | N nitrate | 4.5±0.2Ba | 157.1±10.9 Aa | 33.4±1.0Aa | 21.0±0.8 Ba | 49.7±2.6 Ba | 1475.2±53.2 Aa | 9.9±1.4Aa | 1.6±0.1 Ab |
| June | CK | 4.0±0.9Ab | 83.2±13.0A b | 37.2±1.6Aa | 28.6±2.5 Aa | 77.0±4.7 Aa | 1030.3±41.2 Ab | 7.7±1.2Aa | 1.4±0.1 Ab |
| | N ammonium | 2.2±0.1A Bc | 70.6±0.9Ab | 25.9±1.8Ba | 17.9±0.2 Bb | 31.8±1.7 Bb | 848.5±62.1B b | 4.0±0.0Bb | 0.9±0.1B b |
| | N nitrate | 1.7±0.3Bb | 89.4±10.3A b | 28.7±1.2Bb | 19.8±0.2 Ba | 25.7±0.6 Bb | 667.8±26.5C b | 4.8±0.9A Bb | 1.2±0.1 Ab |
| October | CK | 3.7±0.4Ab | 89.1±0.9Ab | 15.2±0.4A Bb | 9.7±0.3A b | 44.7±0.2 Ab | 578.0±38.1A c | 2.9±0.2Ab | 7.6±0.1 Aa |
| | N ammonium | 3.7±0.1Ab | 64.0±4.2Ab | 16.2±0.9Ab | 5.2±0.1B c | 26.5±0.2 Bb | 423.4±1.6Bc | 2.8±0.1Ab | 5.5±0.8 Aa |
| | N nitrate | 2.2±0.0Bb | 68.3±11.5A b | 13.5±0.1Bc | 5.3±0.1B b | 24.5±0.2 Cb | 409.8±4.7Bc | 1.9±0.1Bc | 5.6±0.8 Aa |

**Supplementary materials**

[Figure]

Fig S1. Average monthly atmospheric temperature and precipitation at the study site during 2015.

**Table S 1** Enzymes and their corresponding substrates and functions.

| Enzyme | Ec | Abbreviation | Substrate | Function |
|---|---|---|---|---|
| Peroxidase | 1.11.1.7 | PER | L-DOPA | Oxidize lignin and aromatic compounds using $H_2O_2$ or secondary oxidants as an electron acceptor |
| Phenol oxidase | 1.10.3.2 | PPO | L-DOPA | Oxidize phenolic compounds using oxygen as an electron acceptor |
| α-1,4-glucosidase | 3.2.1.20 | aG | 4-MUB-α-D-glucoside | Releases glucose from starch |
| β-1,4-glucosidase | 3.2.1.21 | BG | 4-MUB-β-D-glucoside | Releases glucose from cellulose |
| Cellobiohydrolase | 3.2.1.91 | CBH | 4-MUB-β-D-cellobioside | Releases disaccharides from cellulose |
| β-1,4-xylosidase | 3.2.1.37 | BX | 4-MUB-β-D-xyloside | Releases xylose from hemicellulose |
| β-1,4-N-acetylglucosaminidase | 3.2.1.14 | NAG | 4-MUB-N-acetyl-β-D-glucosaminide | Releases N-acetyl glucosamine from oligosaccharides |
| Acid phosphatase | 3.1.3.1 | AP | 4-MUB-phosphate | Releases phosphate groups |

**Table S2** Time-independent seasonal variations in ammonium and PLFAs. Small letters represent significant differences between the sampling time ($P$ <0.05), error bars represent means $\pm$ standard errors (n=9).

| Months | Ammonium mg kg$^{-1}$ | Total PLFA nmol g$^{-1}$ | Bacteria nmol g$^{-1}$ | G$^{-}$ nmol g$^{-1}$ |
|---------|---------|---------|---------|---------|
| March | 23.5±1.0a | 9.2±0.2c | 7.1±0.2c | 2.5±0.1c |
| June | 10.6±1.0b | 11.0±0.2b | 7.7±0.2b | 3.1±0.1b |
| October | 7.5±1.0b | 16.7±0.2a | 13.8±0.2a | 5.0±0.1a |

---

## Short Comment (SC2) · 1 Aug 2017

Comments in response to Referee 2 This manuscript by Zhang et al examines the impact of 4 years of simulated atmospheric N deposition in 30 year old pine-dominated forest plots. The approach used by the authors is novel in that it differentiates between

type of N applied (NH4 vs NO3), which is particularly interesting due to the global stabilization/decline in N deposition as NO3 and the increase in N deposition as NH4.

Response: We would like to thank you for the helpful and constructive comments, which further improved the manuscript. We have carefully revised our manuscript to take account of your comments and suggestions. Please find below our responses to comments. The page and line numbers mentioned here refer to the latest revision of our manuscript simultaneously submitted with all figures as a single PDF file.

General Comments: 1) In general, the data are much more complex across time than the authors present. It would be nice if the results are as clean as suggested in the topic sentence of each discussion paragraph, but it is simply not the case because many of the results are time-dependent. For this reason, the authors need to greatly expand the interpretation and discussion of the treatment x time interaction that is presented in Table 2. Further, to help the reader reason through the data, I think it would be beneficial to collapse the data to not include the 3 sampling times for those factors that do not exhibit a significant treatment x time interaction. For example, Fig 2(i) can be reduced to three bars for control, ammonium N, and nitrate N because there was not a significant interaction.

Response: We discussed the possible reasons resulted in the interaction effects at the discussion section. "The N treatments also varied significantly on a seasonal basis and there were interaction effects between N treatments and seasons on the contents of some PLFA biomarkers and enzyme activities (Table 2). Climate conditions, plant growth, the amount of litter returned, and plant-soil-microorganism systems varied across the three seasons. The temperature ranged from 13.5 to 27.6 °C, and precipitation ranged from 88.2 to 176.6 mm, across the three seasons (Fig. S1), and did not limit the growth of microorganisms. The positive relationships between PLFA biomarker contents and soil moisture contents indicate that soil moisture had a strong influence on soil microbial community biomass. There may be interaction effects between plant growth, the mass and quality of litter, plant-microbe competition, and soil nutrient dynamics. For example, compared with the control plots, the soil DOC contents were lower, and soil nitrate contents stayed the same in June (the growing season) in the ammonium treatment, but the soil DOC and nitrate contents were higher in the ammonium and nitrate treatments in March and October (the non-growing season, Fig. 2). This indicates that there was stronger competition between plants and microbes for available C and N in June than in March and October, and that there were interaction effects between plants and microbes on soil C and N availability. This might explain the interaction effects between N additions and seasons on the activities of C and N-acquisition enzymes. The effects of interactions between N additions and season on the AMF PLFA contents, along with available C and N dynamics, may result from plant growth as plant-AMF symbiotic systems may be influenced by fine root biomass." (line 354-371)

We also simplified the figure that the indexes were treatments-independent, and the figures were shown at Fig. 1and Fig. 3.

2) More can be done with soil enzyme data to forward the authors main hypotheses and ideas that are introduced in line 119. For example, enzyme data can be presented as ratios of C acquiring/N acquiring and/or C acquiring/P acquiring. Such analysis will provide a clearer avenue to draw conclusions about whether microbes can alter how resources are allocated to scavenge for nutrients under different conditions.

Response: "We compared the stoichiometry of C and P to N-acquisition enzyme activities by ln(aG+BG+CBH+BX) and lnaP to lnNAG, respectively (n=27)." (line 210-212) "When compared to control, the ratios of C to N-acquisition enzyme activities were about 0.2 higher, the ratios of N to P acquisition enzyme activities were about 0.1 lower, and there were no obvious differences in the ratios of C to P acquisition enzyme activities in the ammonium and nitrate treatments." (line 277-280) "The ratios of C or P to N acquisition enzyme activities were higher in the ammonium and nitrate treatments than in the control plots, and the N-acquisition enzyme activities per unit of microbial biomass were lower in the ammonium and nitrate treatments than in the control (Fig.

5), indicating that microorganisms secreted enzymes in line with the economic theory. Measured absolute enzyme activities were positively correlated with soil pH and ammonium contents, and negatively correlated with nitrate contents (Fig. 6). The inhibitory effects of N on the soil absolute enzyme activities may be more closely related to abiotic factors, i.e. soil pH and nitrification, than biotic factors (Kivlin et al., 2016)." (line 339-346)

3) Is there any ecological rationale for the March/June/October time points? What is the climatic variation across these times? Also, I assume soil moisture was measured, and if it was, those data should be presented and included in all analyses (including the RDA). Soil moisture has been shown to be a major driver of microbial community composition.

Response: "We collected soil samples in March, June, and October of 2015, to represent spring, summer, and fall." (line 169-170) "The atmospheric conditions and plant-derived litters differed between the three seasons, and so indirectly affected the soil microbial biomass and enzyme activities of different communities. We collected soils from three seasons so that we could investigate the synthetic responses of soil microbial biomass and enzyme activities to ammonium and nitrate additions and to obtain improved information to support predictions of the effects of elevated N depositions on C, N, and P cycling." (line 172-177) "Soil water contents (SWC) were measured by the oven drying method (105 °C)." (line 184-185) "SWC were positively correlated with soil PLFA biomarker contents, but were not correlated with the absolute enzyme activities (Fig. 6)." (line 302-303) "The temperature ranged from 13.5 to 27.6 °C, and precipitation ranged from 88.2 to 176.6 mm, across the three seasons (Fig. S1), and did not limit the growth of microorganisms. The positive relationships between PLFA biomarker contents and soil moisture contents indicate that soil moisture had a strong influence on soil microbial community biomass." (line 357-360) The data applied to RDA analysis included soil moisture contents, and the figure was shown at Fig.6. Average monthly atmospheric temperature and precipitation at the study site during 2015

were shown at Fig. S1.

4) The manuscript needs to be edited for grammar, flow, and word choice. The writing is poor and must be improved significantly in order to be publishable.

Response: We have our revised version manuscript professionally edited by a native English speaker colleague, Dr Deborah Ballantine from the United International College, Beijing Normal University and Hong Kong Baptist University, Zhuhai, Guangdong Province.

Minor comments:

Line 169: Is there rationale for the dose of N applied? Any relation to predictions for future N deposition in the region?

Response: "Background atmospheric wet N deposition of about 33 kg N ha$-1$ yr$-1$ comprises 11 kg N ha$-1$ yr$-1$ as ammonium and 8 kg N ha$-1$ yr$-1$ as nitrate (Zhu et al., 2014). We established a control and test plots at the experimental sites. We equally added two types of N to the test plots, i.e. ammonium (Nammonium) as ammonium chloride (NH4Cl) and nitrate (Nnitrate) as sodium nitrate (NaNO3), at an annual rate of 40 kg N ha$-1$ yr$-1$. This rate was about double the background N wet deposition." (line 154-159)

Line 229: How were the 3 sampling times considered for the RDA? Was the RDA ran on data from one of the three sampling times? Or from average data across the sampling times? Given the treatment by time interaction, this point needs clarification.

Response: The response of soil biomass of different microbial communities and enzyme activities to N treatments was similar in the three sampling seasons, so all of data in the treatments and seasons (n=27) was applied to RDA analysis. And we added the n value to statistical analyses section. (line 231)

Line 315: Others have shown that N addition disproportionately effects soil fungi and may stimulate soil bacteria (for example, see doi 10.3389/fmicb.2016.00259 and

10.1128/AEM.01224-14). This dynamic may also help explain the increase in DOC observed with N addition.

Response: We have added that " Moreover, the higher soil DOC concentrations observed in the nitrate-addition treatments (Fig. 2) may be attributed to changes in the diversity of the composition of saprophytic bacteria (Freedman and Zak, 2014; Freedman et al., 2016)." (line 324-326)

Table 2: I think P-values can be removed from this table. Given that significant values are bolded, P-values are redundant and make the table busy for the reader.

Response: Revised as recommended, please refer to Table 1.

Please also note the supplement to this comment:
https://www.biogeosciences-discuss.net/bg-2017-179/bg-2017-179-SC2-supplement.pdf

―――――――――――――――――――――

[revised manuscript text omitted]

---

## Author Comment (AC1) · 11 Aug 2017

We are grateful for the reviewers comments and we agree to substantial changes to the manuscript in response to the valuable input from the two reviewers. Please find our response to reviewers comments on behalf of all co-authors. Our response to the reviewer's comments are detailed point by point below.

Yours sincerely

Chuang Zhang, Xin-Yu Zhang, Hong-Tao Zou, Liang Kou, Yang Yang, Xue-Fa Wen,

[Figure]

Sheng-Gong Li, Hui-Min Wang, Xiao-Min Sun

Response to reviewers comments

RC1 Anonymous referee #1

1. The data analyses is not enough or suitable. 1) The authors mentioned that using two factors randomized block variance of analyses (ANOVA) to test the differences between the treatment and the sampling season. However, after reading the whole manuscript, I did not find any results from this methods. For example, if there is two factors, it must have the possible interaction effect between two factors. Actually, from Table 2, I have found several important interaction effects for DOC, Nitrate, Fugal et al. However, the authors did not discuss this at all.

Response: We added the results of interaction effects to the result section. "The soil pH and ammonium contents were either treatment- or time-independent. There were interaction effects between the treatments and the sampling time on the soil DOC and nitrate contents (P<0.01, Table 1)." (line 238-240) "Both the treatment and the time of sampling significantly influenced the soil microbial biomass of the different communities (P<0.01). Total PLFAs, bacteria, G−, and G+/G− were either treatment- or time-independent. There were also interaction effects between treatments on sampling time and fungi, actinomycetes, G+, AMF, SAP, and the fungi/bacteria ratio (Table 1)." (line 252-255) "There were significant influences from both treatment and sampling time on the measured absolute enzyme activities (P<0.01). Activities of BG, AP, and PPO were either treatment- or time-independent, and there were interaction effects between the treatments and sampling time on activities of aG, BX, CBH, NAG, and PER (Table 1)." (line 271-274) We also discussed the possible reasons resulted in the interaction effects at the discussion section. "The N treatments also varied significantly on a seasonal basis and there were interaction effects between N treatments and seasons on the contents of some PLFA biomarkers and enzyme activities (Table 2). Climate conditions, plant growth, the amount of litter returned, and plant-soil-microorganism systems

varied across the three seasons. The temperature ranged from 13.5 to 27.6 °C, and precipitation ranged from 88.2 to 176.6 mm, across the three seasons (Fig. S1), and did not limit the growth of microorganisms. The positive relationships between PLFA biomarker contents and soil moisture contents indicate that soil moisture had a strong influence on soil microbial community biomass. There may be interaction effects between plant growth, the mass and quality of litter, plant-microbe competition, and soil nutrient dynamics. For example, compared with the control plots, the soil DOC contents were lower, and soil nitrate contents stayed the same in June (the growing season) in the ammonium treatment, but the soil DOC and nitrate contents were higher in the ammonium and nitrate treatments in March and October (the non-growing season, Fig. 2). This indicates that there was stronger competition between plants and microbes for available C and N in June than in March and October, and that there were interaction effects between plants and microbes on soil C and N availability. This might explain the interaction effects between N additions and seasons on the activities of C and N-acquisition enzymes. The effects of interactions between N additions and season on the AMF PLFA contents, along with available C and N dynamics, may result from plant growth as plant-AMF symbiotic systems may be influenced by fine root biomass." (line 354-371)

2) Another question is that I have some confused why the authors took three measurements for PLFA biomarkers, even we knew that the variance of PLFA measurements varied a lot. Is there any important reasons to choose these three different months and what the ecological meanings are? evenly sprayed onto the plot once per month. Did this mean that the frequency of spray the nitrogen 12 times per year?

Response: "We collected soil samples in March, June, and October of 2015, to represent spring, summer, and fall." (line 169-170) "The atmospheric conditions and plant-derived litters differed between the three seasons, and so indirectly affected the soil microbial biomass and enzyme activities of different communities. We collected soils from three seasons so that we could investigate the synthetic responses of soil micro-

bial biomass and enzyme activities to ammonium and nitrate additions and to obtain improved information to support predictions of the effects of elevated N depositions on C, N, and P cycling." (line 172-177) The frequency of spray the nitrogen was 12 times per year. We showed at "The NH4Cl or NaNO3 were dissolved in 30 L of tap water and evenly sprayed onto the plots once a month, i.e. 12 times per year." (line 161-162)

Other minor comments:

1. In introduction and discussion section: the authors cited papers in different ways, please format it.

Response:Revised as recommended.

2. The authors should improve their English grammar for a lot. There are a lot of small mistakes in the whole manuscript.

Response: We have our revised version manuscript professionally edited by a native English speaker colleague, Dr Deborah Ballantine from the United International College, Beijing Normal University and Hong Kong Baptist University, Zhuhai, Guangdong Province.

RC1 Anonymous referee #2

General Comments:

1) In general, the data are much more complex across time than the authors present. It would be nice if the results are as clean as suggested in the topic sentence of each discussion paragraph, but it is simply not the case because many of the results are time-dependent. For this reason, the authors need to greatly expand the interpretation and discussion of the treatment x time interaction that is presented in Table 2. Further, to help the reader reason through the data, I think it would be beneficial to collapse the data to not include the 3 sampling times for those factors that do not exhibit a significant treatment x time interaction. For example, Fig 2(i) can be reduced to three bars for control, ammonium N, and nitrate N because there was not a significant interaction.

Response: We discussed the possible reasons resulted in the interaction effects at the discussion section. "The N treatments also varied significantly on a seasonal basis and there were interaction effects between N treatments and seasons on the contents of some PLFA biomarkers and enzyme activities (Table 2). Climate conditions, plant growth, the amount of litter returned, and plant-soil-microorganism systems varied across the three seasons. The temperature ranged from 13.5 to 27.6 °C, and precipitation ranged from 88.2 to 176.6 mm, across the three seasons (Fig. S1), and did not limit the growth of microorganisms. The positive relationships between PLFA biomarker contents and soil moisture contents indicate that soil moisture had a strong influence on soil microbial community biomass. There may be interaction effects between plant growth, the mass and quality of litter, plant-microbe competition, and soil nutrient dynamics. For example, compared with the control plots, the soil DOC contents were lower, and soil nitrate contents stayed the same in June (the growing season) in the ammonium treatment, but the soil DOC and nitrate contents were higher in the ammonium and nitrate treatments in March and October (the non-growing season, Fig. 2). This indicates that there was stronger competition between plants and microbes for available C and N in June than in March and October, and that there were interaction effects between plants and microbes on soil C and N availability. This might explain the interaction effects between N additions and seasons on the activities of C and N-acquisition enzymes. The effects of interactions between N additions and season on the AMF PLFA contents, along with available C and N dynamics, may result from plant growth as plant-AMF symbiotic systems may be influenced by fine root biomass." (line 354-371) We also simplified the figure that the indexes were treatments-independent, and the figures were shown at Fig. 1and Fig. 3.

2) More can be done with soil enzyme data to forward the authors main hypotheses and ideas that are introduced in line 119. For example, enzyme data can be presented as ratios of C acquiring/N acquiring and/or C acquiring/P acquiring. Such analysis will provide a clearer avenue to draw conclusions about whether microbes can alter how resources are allocated to scavenge for nutrients under different conditions.

Response: "We compared the stoichiometry of C and P to N-acquisition enzyme activities by ln(aG+BG+CBH+BX) and lnaP to lnNAG, respectively (n=27)." (line 210-212) "When compared to control, the ratios of C to N-acquisition enzyme activities were about 0.2 higher, the ratios of N to P acquisition enzyme activities were about 0.1 lower, and there were no obvious differences in the ratios of C to P acquisition enzyme activities in the ammonium and nitrate treatments." (line 277-280) "The ratios of C or P to N acquisition enzyme activities were higher in the ammonium and nitrate treatments than in the control plots, and the N-acquisition enzyme activities per unit of microbial biomass were lower in the ammonium and nitrate treatments than in the control (Fig. 5), indicating that microorganisms secreted enzymes in line with the economic theory. Measured absolute enzyme activities were positively correlated with soil pH and ammonium contents, and negatively correlated with nitrate contents (Fig. 6). The inhibitory effects of N on the soil absolute enzyme activities may be more closely related to abiotic factors, i.e. soil pH and nitrification, than biotic factors (Kivlin et al., 2016)." (line 339-346)

3) Is there any ecological rationale for the March/June/October time points? What is the climatic variation across these times? Also, I assume soil moisture was measured, and if it was, those data should be presented and included in all analyses (including the RDA). Soil moisture has been shown to be a major driver of microbial community composition.

Response: "We collected soil samples in March, June, and October of 2015, to represent spring, summer, and fall." (line 169-170) "The atmospheric conditions and plant-derived litters differed between the three seasons, and so indirectly affected the soil microbial biomass and enzyme activities of different communities. We collected soils from three seasons so that we could investigate the synthetic responses of soil microbial biomass and enzyme activities to ammonium and nitrate additions and to obtain improved information to support predictions of the effects of elevated N depositions on C, N, and P cycling." (line 172-177) "Soil water contents (SWC) were measured

by the oven drying method (105 °C)." (line 184-185) "SWC were positively correlated with soil PLFA biomarker contents, but were not correlated with the absolute enzyme activities (Fig. 6)." (line 302-303) "The temperature ranged from 13.5 to 27.6 °C, and precipitation ranged from 88.2 to 176.6 mm, across the three seasons (Fig. S1), and did not limit the growth of microorganisms. The positive relationships between PLFA biomarker contents and soil moisture contents indicate that soil moisture had a strong influence on soil microbial community biomass." (line 357-360) The data applied to RDA analysis included soil moisture contents, and the figure was shown at Fig.6. Average monthly atmospheric temperature and precipitation at the study site during 2015 were shown at Fig. S1.

4) The manuscript needs to be edited for grammar, flow, and word choice. The writing is poor and must be improved significantly in order to be publishable.

Response: We have our revised version manuscript professionally edited by a native English speaker colleague, Dr Deborah Ballantine from the United International College, Beijing Normal University and Hong Kong Baptist University, Zhuhai, Guangdong Province.

Minor comments: Line 169: Is there rationale for the dose of N applied? Any relation to predictions for future N deposition in the region?

Response: "Background atmospheric wet N deposition of about 33 kg N ha−1 yr−1 comprises 11 kg N ha−1 yr−1 as ammonium and 8 kg N ha−1 yr−1 as nitrate (Zhu et al., 2014). We established a control and test plots at the experimental sites. We equally added two types of N to the test plots, i.e. ammonium (Nammonium) as ammonium chloride (NH4Cl) and nitrate (Nnitrate) as sodium nitrate (NaNO3), at an annual rate of 40 kg N ha−1 yr−1. This rate was about double the background N wet deposition." (line 154-159)

Line 229: How were the 3 sampling times considered for the RDA? Was the RDA ran on data from one of the three sampling times? Or from average data across the sampling

times? Given the treatment by time interaction, this point needs clarification.

Response: The response of soil biomass of different microbial communities and enzyme activities to N treatments was similar in the three sampling seasons, so all of data in the treatments and seasons (n=27) was applied to RDA analysis. And we added the n value to statistical analyses section. (line 231)

Line 315: Others have shown that N addition disproportionately effects soil fungi and may stimulate soil bacteria (for example, see doi 10.3389/fmicb.2016.00259 and 10.1128/AEM.01224-14). This dynamic may also help explain the increase in DOC observed with N addition.

Response: We have added that " Moreover, the higher soil DOC concentrations observed in the nitrate-addition treatments (Fig. 2) may be attributed to changes in the diversity of the composition of saprophytic bacteria (Freedman and Zak, 2014; Freedman et al., 2016)." (line 324-326)

Table 2: I think P-values can be removed from this table. Given that significant values are bolded, P-values are redundant and make the table busy for the reader.

Response: Revised as recommended, please refer to Table 1.

Please also note the supplement to this comment:
https://www.biogeosciences-discuss.net/bg-2017-179/bg-2017-179-AC1-supplement.pdf

---

## Author Response (AR1)

**The response to comments**

Dear Dr. Denise M. Akob

We are grateful for your comments and we agree to revise the manuscript in response to the comments. Please find our response to the comments on behalf of all co-authors. Our response to the reviewer's comments are detailed point by point below.

Abstract: there is no mention of slash pine plantation but it is part of the title. Please revise to include and to link your results back to the habitat or if the habitat is not the key part of the story then revise the title.

Response: We accepted your suggestion. And we remove the "slash pine" from the title.

L. 33: change to hydrolysis. The correct term is hydrolysis and cannot be plural

Response: Revised as recommended, please refer to Line 34.

L. 50: change to "is mostly comprised of"

Response: Revised as recommended, please refer to Line 51.

L. 69: define the BG and NAG acronyms

Response: BG is β-1,4-glucosidase, and NAG is β-1,4-N-acetylglucosaminidase. We add the full names to the Line 70. And at the same time, we used the abbreviations of βG and NAG at Line 97-98.

L. 106: "in line with the economic theory" does not fit here. No economic theory has been presented. I would omit it and start with "Microorganisms…" or try to introduce this better

Response: We accepted your suggestion. The sentence has been revised as 'Microorganisms will allocate energy to the relatively absent resources, so that N additions will cause C and P-acquisition enzymes to increase, and N-acquisition enzymes to decrease (Burns et al., 2013)'.  (Line 108-110).

L. 114-116: add a reference

Response: The reference of 'Sinsabaugh et al., 2002' has been added (Line 119).

L. 134: change to "C-"

Response: Revised as recommended, please refer to Line 137.

L. 135: change to ";"

Response: Revised as recommended, please refer to Line 138.

L. 142: there are so many acronyms in the paper that I would not abbreviate the experimental station, just write it out.

Response: The 'QYZ' has been revised as Qianyanzhou (Line 145).

L. 149: define a.s.l.

Response: a.s.l means above sea level (Line 146).

L. 153-165: this is still not clear. How many test plots were there, first you say one control and one test then you state there are 9? Start with how they were divided up then discuss the treatments. On l. 157, it reads as if you added both NH4 and NO3 to both test plots. So, how were you comparing effects of each N source? It might be useful to add a schematic to supplemental information. For the application, how much time was there between sprayings? It was only 1 day per month when N was applied?

Response: We have revised the sentence as 'Nine $20 \times 20$ m plots were established at the experimental sites, including a control, ammonium only and nitrate only additions plots with three replicates (3 treatments $\times$ 3 replicates).'. Furthermore, a schematic was added as supplemental figure S2. The N were applied on a non-rainy day at the interval of about one month, so the N was applied one day per month (Line 168-169).

L. 174: I would not use synthetic here. I suggest changing to seasonal.

Response: Revised 'synthetic' to 'seasonal' (Line 178).

L. 183-189: was the soil:water shaken before measurement of pH and N? In this section, either provide details on the methods or provide references.

Response: The homogenate was stirring by glass rod for one minute and then was settled for 30 min before measurement of soil pH. And the soil-water mixture was shaken for 2 h before measurement of soil ammonium, nitrate, soil dissolved organic carbon. We have added the references [Bao. (2010)] in the manuscript (Line 187)

L. 187: soil cannot be extracted with soil, revise

Response: The sentence has been revised as 'Soil DOC was extracted with distilled water at a ratio of 1 g soil : 5 ml water, and was measured with an organic element analyzer (Liquid TOCII, Elementar, Germany)' (Line 192-195)

L. 205: what concentration of sodium acetate?

Response: The concentration of sodium acetate was 50 mmol L$^{-1}$ (Line 212).

L. 241: specify panel A, e.g., Fig 1A

Response: We have specified the panel (a) to (f) for Figure 1-6. Accordingly, we have revised in the results and discussion.

L. 241-242: incorrect usage of "respectively". Please always refer to the specific figure panels you are referencing.

Response: We have revised the sentence as "The soil nitrate contents were 165% and 129% higher (Fig. 2b), and the soil ammonium contents were 31% and 38% lower in the ammonium and nitrate treatments (Fig. 1b) than in the control for the three sampling events." (Line 248-251).

L. 258: you are missing an "and"

Response: We added it to Line 265.

L. 264: I caution against using the term "shifted" when you are not showing or referring to time series data.

Response: We have revised the sentence as "The microbial communities was dominated by G$^+$ in the ammonium-treated plots, meaning that the G$^+$/G$^-$ ratios were higher in the ammonium-treated plots than in the control or nitrate-treated plots." (Line 272-274)

L. 273-283, Table 2 and elsewhere: are the enzymes supposed to be named with Greek letters? The naming is inconsistent with the intro and methods. Please verify and correct throughout the paper.

Response: We revised aG, BG, BX to αG, βG, βX throughout the manuscript, the Tables and the Figures.

L. 292-294: this sentence is strangely worded—it is unclear what you are referring to with use of respectively twice in this sentence. What are the values respective of?

Response: We revised the sentence to "The results of RDA between soil properties and absolute enzyme activities showed that the first axis explained 72.0% of the variability (Fig. 6a), while the results of RDA between soil properties and microbial community structures showed that the first axis explained 67.5% of the variability (Fig. 6b)." (Line 301-308)

L. 292-303: refer to your table and figure earlier in the paragraph

Response: We added "Fig.6a" to Line 302, and added "Fig. 6b" to Line 303, and added "Fig.6a,b" to Line 311 and Line 317.

L. 312-313: C cannot be included in the sum of ammonium and nitrate, remove the parenthetical statement

Response: We removed the parenthetical statement. Please refer to Line 326-327.

L. 350: only include the reference once.

Response: We removed "(Li et al., 2016)" from Line 365.

L. 356-357: do you mean microbial communities? Microorganism systems is an unsual choice.

Response: The interaction effects between plant, microbes and soil nutrients were variable across the three sampling events. So we changed the "plant-soil-microorganism systems" to "plant-microorganism competitive relationship", please refer to Line 372.

L. 560: Bacteria is spelled wrong

Response: Revised as recommended. Please refer to Fig.3b

L. 558: soil dissolved organic carbon is misspelled

Response: Fig.2a was revised as recommended.

Fig. 4f: the "Cb" is not easy to read

Response: Fig.4f was revised as recommended. Capital letters represent significant differences between the treatments (P <0.05), and small letters represent significant differences between the sampling time (P <0.05).

Fig. 6: please define the abbreviations in the legend

Response: The full names of the PLFA biomarkers, enzymes and soil properties were shown in Table 1. We add 'The abbreviations are the same as Table 1. SOC: soil organic matter; TN: total nitrogen; C/N: the ratio of soil organic matter to total nitrogen; SWC: soil water contents.' to the legend of Fig.6. At the same time, we omitted the state of the abbreviations shown in the legend of Fig. 3 and 4 that replicated with Table 1.

**The list of all relevant changes made in the manuscript**

The line number see the marked-up manuscript that was showed below this section.

1. Title: Delete 'in a slash pine plantation'

2. Line 33-34: Change 'hydrolyses' to 'hydrolysis'

3. Line 51: Change 'comprises' to 'is comprised of'

4. Line 70: Change 'BG and NAG' to 'β-1,4-glucosidase (βG) and β-1,4-N-acetylglucosaminidase (NAG)'

5. Line 97-98: Change 'β-1,4-glucosidase (βG)' and 'β-1,4-N-acetylglucosaminidase (NAG)' to 'βG' and 'NAG'

6. Line 108-111: Revised as 'Microorganisms will allocate energy to the relatively absent resources so that N additions will cause C and P-acquisition enzymes to increase, and N-acquisition enzymes to decrease (Burns et al., 2013).'

7. Line 119: Add '(Sinsabaugh et al., 2002)'

8. Line 137: Change 'C and P-hydrolase' to 'C- and P-hydrolase'

9. Line 138: Change ',' to ';'

10. Line 145: delete the abbreviation '(QYZ)'

11. Line 146-147: Change 'a.s.l' to 'above sea level'

12. Line 158-165: Revised as 'Nine $20 \times 20$ m plots were established at the experimental sites, including a control, ammonium only and nitrate only treatments with three replicates (3 treatments $\times$ 3 replicates).'

13. Line 178: Change 'synthetic' to 'seasonal'

14. Line 187: We added 'The measurement of soil chemical properties was followed the method of Bao (2010).'

15. Line 272-274: Revised as 'The microbial communities shifted from G− towas dominated by G+ in the ammonium-treated plots, meaning that the G+/G− ratios were higher in the ammonium-treated plots than in the control or nitrate-treated plots (Fig. 3d).'

16. Line 301-308: Revised as 'The results of RDA between soil properties and absolute enzyme activities showed that the first axis explained 72.0% of the variability (Fig. 6a), while the results of RDA between soil properties and microbial community structures showed that the first axis explained 67.5% of the variability (Fig. 6b).'

17. Line 326-327: Delete '(the sum of the ammonium and nitrate concentrationscontents)'

18. Results and Discussion section: We have specified the panel (a) to (f) for Figure 1-6.

19. We revised aG, BG, BX to αG, βG, βX throughout the manuscript, the Tables and the Figures.

20. We added a schematic as supplemental figure S2.

**The marked-up manuscript see below**

[revised manuscript text omitted]

**Supplementary materials**

[Figure]

**Fig S1.** Average monthly atmospheric temperature and precipitation at the study site during 2015.

[Figure]

| Treatments | N form | N dose |
|---|---|---|
| CK | -- | 0 kg N ha$^{-1}$ yr$^{-1}$ |
| N$_{ammonium}$ | NH$_4$Cl | 40 kg N ha$^{-1}$ yr$^{-1}$ |
| N$_{nitrate}$ | NaNO$_3$ | 40 kg N ha$^{-1}$ yr$^{-1}$ |

**Fig S2. The schematic diagram of the experimental treatments.**

**Table S 1** Enzymes and their corresponding substrates and functions.

| Enzyme | Ec | Abbreviation | Substrate | Function |
|---|---|---|---|---|
| Peroxidase | 1.11.1.7 | PER | L-DOPA | Oxidize lignin and aromatic compounds using $H_2O_2$ or secondary oxidants as an electron acceptor |
| Phenol oxidase | 1.10.3.2 | PPO | L-DOPA | Oxidize phenolic compounds using oxygen as an electron acceptor |
| α-1,4-glucosidase | 3.2.1.20 |  | 4-MUB-α-D-glucoside | Releases glucose from starch |
| β-1,4-glucosidase | 3.2.1.21 |  | 4-MUB-β-D-glucoside | Releases glucose from cellulose |
| Cellobiohydrolase | 3.2.1.91 | CBH | 4-MUB-β-D-cellobioside | Releases disaccharides from cellulose |
| β-1,4-xylosidase | 3.2.1.37 |  | 4-MUB-β-D-xyloside | Releases xylose from hemicellulose |
| β-1,4-N-acetylglucosaminidase | 3.2.1.14 | NAG | 4-MUB-N-acetyl-β-D-glucosaminide | Releases N-acetyl glucosamine from oligosaccharides |
| Acid phosphatase | 3.1.3.1 | AP | 4-MUB-phosphate | Releases phosphate groups |

**Table S2** Time-independent seasonal variations in ammonium and PLFAs. Small letters represent significant differences between the sampling time ($P < 0.05$), error bars represent means ± standard errors (n=9).

| Months | Ammonium | Total PLFA | Bacteria | G⁻ |
|---|---|---|---|---|
| | mg kg$^{-1}$ | nmol g$^{-1}$ | nmol g$^{-1}$ | nmol g$^{-1}$ |
| March | 23.5±1.0a | 9.2±0.2c | 7.1±0.2c | 2.5±0.1c |
| June | 10.6±1.0b | 11.0±0.2b | 7.7±0.2b | 3.1±0.1b |
| October | 7.5±1.0b | 16.7±0.2a | 13.8±0.2a | 5.0±0.1a |

---

## Author Response (AR2)

Dear Dr. Denise M. Akob

We are grateful for your comments and we agree to revise the manuscript in response to the comments.

Please find our response to the comments on behalf of all co-authors. Our response to the reviewer's comments are detailed point by point below.

Associate Editor's Comments

I am selecting publish subject to technical corrections as Figure S2 is of poor quality. Please revising and upload a better version for final publication.

Response: We revised the Fig.S2, please refer to Fig.S2 in the supplementary materials.

Additionally, we added "“Key Laboratory of Agricultural Water Resources, Center for Agricultural Resources Research, Institute of Genetics and Developmental Biology, Chinese Academy of Sciences, 286 Huaizhong Road, Shijiazhuang 050021, China" to Zhang Chuang (First author).